## TOPICAL REVIEW

# Astrocytes: Orchestrators of brain gas exchange and oxygen homeostasis

Isabel N. Christie 

*School of Medicine and Population Health, University of Sheffield, SHEFFIELD, UK*

Handling Editors: Laura Bennet & Nathan Schoppa

The peer review history is available in the Supporting Information section of this article (https://doi.org/10.1113/JP288934#support-information-section).

**Abstract figure legend** Oxygen and carbon dioxide enter the body via breathing; in the brain astrocytes play a key role balancing oxygen delivery with carbon dioxide removal.

**Abstract** If we consider neurons like muscles during exercise, the demand for oxygen ($O_2$) and carbon dioxide ($CO_2$) elimination is constantly changing. This review summarises the evidence that astrocytes are essential for homeostasis of the respiratory gases in the brain, with a particular focus on oxygen homeostasis. Astrocytes surround cerebral blood vessels and sense changes in oxygen availability in the milieu. They contribute to pH homeostasis and are increasingly recognized for their contribution to central chemosensitivity, particularly in detecting changes in $CO_2$ and proton ($H^+$) concentrations. They are one of the cell types that govern changes in cerebral perfusion rate. Cerebral perfusion dynamically matches tissue metabolism, to balance $O_2$ delivery and $CO_2$ removal. By examining the role of astrocytes as both sensors and effectors in this homeostatic balancing act, this review argues that astrocytes influence the metabolic environment of neural networks with profound implications for cognitive function.

**Isabel Christie** is a neuroscientist that specialises in pre-clinical imaging, gas exchange, astrocytes and cerebral perfusion. She trained at the Centre for Advanced Biomedical Imaging and undertook postdoctoral research at the Centre for Cardiovascular and Metabolic Neuroscience at University College London. In 2024 she received funding from the Wellcome Trust to start her own lab at the University of Sheffield.

The Journal of Physiology

(Received 24 March 2025; accepted after revision 23 July 2025; first published online 20 August 2025)

**Corresponding author** I. N. Christie: School of Medicine and Population Health, University of Sheffield, Sheffield, UK.
Email: i.christie@sheffield.ac.uk

### Oxygen homeostasis in the brain

The human brain has a high metabolic demand, consuming approximately 20–25% of the body's total energy and 20% of its $O_2$ despite constituting only 2% of total body weight. This organ's metabolic demand necessitates a continuous and precisely regulated supply of $O_2$ and glucose to support metabolic processes and maintain proper functioning. Oxygen is indispensable as the final electron acceptor in oxidative phosphorylation, the primary mechanism for ATP generation in brain cells. Sustained deviations from optimal physiological $O_2$ levels (normoxia) can lead to increased production of reactive oxygen species (ROS) by the electron transport chain, causing oxidative damage, cellular dysfunction, or even cell death. However, the metabolic demands of neural networks vary in space and time. Finite, discreet regions of the brain become more active during various cognitive tasks and additional blood is delivered to these brain regions. This phenomenon, dubbed neurovascular coupling, has been studied for over a century and is frequently exploited for non-invasive brain imaging. Homeostasis is a core principle in physiology in which a dynamic equilibrium occurs (see Fig. 1), within a range of values, often likened to a thermostat. Oxygen homeostasis in the human brain appears to occur over the normoxic range of 25–40 mmHg (Jeffcote et al., 2024, Pennings et al., 2008). This review will present evidence of how astrocytes operate to maintain normoxia and how astrocytes behave when tissue $O_2$ fall below 25 mmHg.

### Summary of evidence that astrocytes maintain oxygen homeostasis

Mathiesen et al. (2012) were first to show spontaneous calcium activity in Bergman glia and that the frequency of such calcium excitability was elevated with age and hypoxia. Recent experimental data show astrocytes detect cerebral hypoxia (Angelova et al., 2015) and release nitric oxide (NO) to maintain $O_2$ homeostasis in the central nervous system (Christie et al., 2023). The brain's demand for $O_2$ delivery and $CO_2$ elimination are constantly changing. By integrating the rate of perfusion (Marina et al., 2020), the balance of $O_2$ (Christie et al., 2023), $CO_2$ (Hosford et al., 2022) and pH (Theparambil et al., 2020), astrocytes influence the metabolic environment of neural networks. When and if astrocytes fail in these roles, neurons struggle to operate. Therefore, astrocytes are essential for gas homeostasis of the mammalian brain with profound implications for cognitive function. With this growing awareness of the significance of astrocytes, biotech and drug companies are turning their attention to the therapeutic potential of targeting these cells.

### Breathing underpins gas homeostasis and astrocytes govern breathing rate

The discovery that astrocytes contribute to the regulation of breathing elevated the importance of glial cells. People breathe faster when they exercise because the metabolic rate of the muscles changes and the respiratory system adapts to eliminate the additional $CO_2$. In parallel, two groups discovered that astrocytes are the primary cells of the medulla responsible for the chemosensitivity that detects both changes in pH ($H^+$) and changes in the circulating partial pressure of $CO_2$ (Ritucci, et al., 2005; Spyer et al., 2004). The subsequent work of Gourine et al. (2005) demonstrated the role of adenosine triphosphate (ATP) as the signalling molecule in this pathway. Purines such as ATP (and its metabolites ADP and AMP) act as chemical signalling molecules and pre-date more complex neuromodulators such as dopamine and serotonin (Verkhratsky & Burnstock, 2014). An excellent review is already in print on the topic of $CO_2$ brain sensing (Gourine & Dale, 2022). This review will focus on $O_2$ and cerebral perfusion, but the reader should keep $CO_2$ in mind since elimination of waste gases is an essential component of gas homeostasis.

### Highlights of recent astrocyte research

Glia are described as non-excitable cells, because they do not conduct action potentials like neurons. Astrocytes have a resting membrane potential of $-60$ to $-85$ mV and can be patched, as neurons are, to measure changes in membrane potential. Astrocytes exhibit rapid and varied fluctuations in the intracellular concentration of unbound calcium $[Ca^{2+}]_i$, with an array of temporal and spatial variability (Bindocci et al., 2017; Semyanov et al., 2020). Multi photon imaging with calcium sensitive dyes is routinely used to study the rodent brain (Denk et al., 1990). Genetically encoded indicators or bulk loaded indicators are targeted to particular cell types and changes in fluorescence are measured as a proxy of subcellular chemical changes. For astrocytes, scientists commonly

study changes in the $[Ca^{2+}]_i$, which at rest is maintained within the nanomolar range (50–10 nM) within the intracellular space (Zheng et al., 2015). Upon $Ca^{2+}$ release from the endoplasmic reticulum via the inositol 1,4,5-trisphosphate ($IP_3$) calcium pathway or entry via calcium channels in the cell membrane, a ∼1000-fold rise in $[Ca^{2+}]_i$ can trigger gliotransmission (Savtchouk & Volterra, 2018), upregulation of transcription factors (Hasel et al., 2021) and other events. The methods devised to capture these data are constantly improving, for example, awake imaging to minimise the confounds of anaesthesia, chronic cranial windows to minimise the confounds of acute surgical inflammation, and genetically encoded indicators to specify the signal is detected from distinct cell types.

To probe the necessity of astrocytes in physiology, scientists have sought to suppress, ablate or inhibit the activity of astrocytes using a variety of methods. Early attempts to block astrocyte activity used a calcium buffer, BAPTA-AM, dialysed into individual astrocytes via a patch pipette; the buffer would diffuse throughout the astrocyte syncytium via gap junctions and suppress calcium dynamics (Enkvist et al., 1989). More recently an $IP_3$ knockout transgenic mouse was generated (Petravicz et al., 2008), targeted to the Gq-coupled metabotropic receptor expressed by astrocytes, to test

the importance of calcium transients that arise from the endoplasmic reticulum. Prior to this, several research groups had characterised astrocyte-mediated changes in vessel diameter, and it was believed $[Ca^{2+}]_i$ events were necessary for changes in cerebral vessel diameter (Mulligan & MacVicar, 2004, Takano et al., 2006, Zonta et al., 2003). There continues to be little consensus about the temporal and functional significance of astrocytic $[Ca^{2+}]_i$ for cerebrovascular dilations. Many publications report calcium activity in endfeet and changes in vessel diameter, but the timing and order of events varies between labs (Lind et al., 2013, Lind et al., 2018, Nizar et al., 2013, Otsu et al., 2015). In 2014, the IP3RKO mouse was used to test the functional necessity of $[Ca^{2+}]_i$ in neurovascular coupling (Bonder & McCarthy, 2014). Data gathered from lightly sedated mice found that vascular responses to visual stimulation were unaffected by the absence of $IP_3$ receptors suggesting that calcium was not necessary (Bonder & McCarthy 2014). However, a more recent paper in awake mice shows that neurovascular coupling responses are affected by knockdown of the $IP_3$ receptor (Lind & Volterra, 2025). This work, conducted using 3D imaging of vessel dilations and astrocyte endfeet, shows that calcium precedes dilation. But notably the authors 'found parameters of the dilation patterns that were unchanged in IP3RKO mice' (Lind

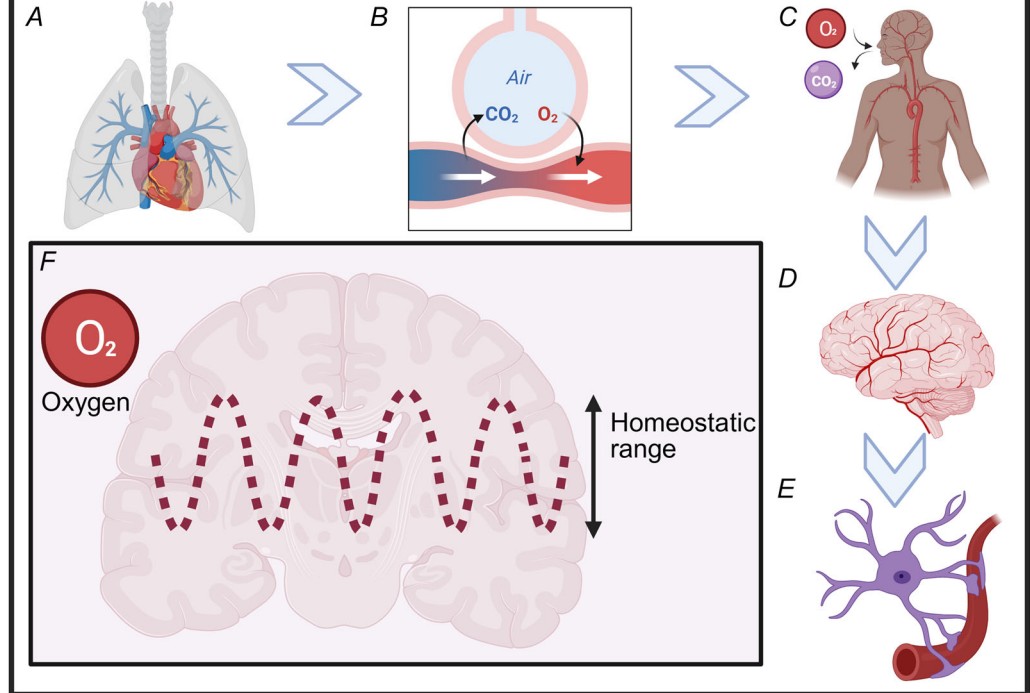

**Figure 1. Schematic illustration of oxygen homeostasis**
*A*, lungs where oxygen enters the body. *B*, gas exchange occurs at the alveoli inside the lungs. *C*, illustration of blood supply to the brain. *D*, an illustration of cerebral vasculature. *E*, an illustration of an astrocyte ensheathing the vessel with endfeet. *F*, an illustration to show the homeostatic range. Created in BioRender. https://BioRender.com/nrlwn9u.

& Volterra, 2025). An interim paper from the Newman lab showed that calcium activity occurs after the onset of dilation and declared no association between astrocyte calcium and cortical capillaries (Del Franco et al., 2022).

Weighing up all of the evidence, I have adopted the language of Tran et al. (2018) to conclude that astrocytes integrate behavioural state and vascular signals. The data from his paper shows the calcium events occur after the dilation, but interestingly the amplitude of the calcium event is affected by the behavioural state (Tran et al., 2018). Earlier work from Grant Gordon showed that the glycolytic state of the brain influences whether astrocytes drive dilation or constriction of cerebral vessels. The groundbreaking paper from the MacVicar lab showed that when hypoxia occurs, astrocyte-mediated vaso-constrictions convert to vasodilations (Gordon et al., 2008). To quote the paper: "Under quiescent periods when $O_2$ is not being rapidly consumed, astrocyte $Ca^{2+}$ signals induce constrictor tone, keeping CBF at an appropriate level. At the onset of neural activity there is a drop in $PO_2$ and rise in extracellular lactate which promotes astrocyte mediated dilation" (Gordon et al., 2008). A more recent paper from the Gordon lab showed that astrocytes amplify the blood flow response to sustained neuronal activity (Institoris et al., 2022). Specifically, this paper shows that the neurovascular coupling response is biphasic, and that astrocyte calcium activity is essential for the second augmentation phase. Perhaps the biphasic signals have some similarities with data observed by Lind and Volterra when they refer to "parameters of the dilation patterns". The amplitude and micro domains of calcium activity within astrocytes may encode and integrate many sources of incoming information as described next.

Most work is now conducted in awake freely moving mice, which means respiratory gases are naturally kept within the normal range and anaesthesia does not suppress brain activity. A significant publication from the Bergles lab linked sensory evoked calcium activity in the visual cortex with noradrenergic activity triggered by a startle (Paukert et al., 2014). The responsivity of the astrocytes to the visual stimulus appeared to be lowered by the presence of noradrenaline, which led the authors to conclude that noradrenaline shifts the gain of astrocyte networks. The idea that astrocytes globally drive changes in cortical state was first proposed in 2011 although at this stage there was no suggestion noradrenaline was involved (Poskanzer & Yuste, 2011). Around the same time research from the Nedergaard lab illustrated noradrenergic modulation of the calcium excitability of cortical astrocytes (Ding et al., 2013). It was clear from these publications that a meaningful inter-action between astrocytes and noradrenergic signalling existed, which was described in an opinion piece which named noradrenaline as the master regulator (O'Donnell et al., 2015). Data from the Hirase lab supported this

view, showing that the interaction of astrocytes and noradrenergic modulation caused changes in learning and memory (Oe et al., 2020). Recently functional magnetic resonance imaging (MRI) data acquired from mice shows that activity of the locus coeruleus and subsequent noradrenaline release cause state changes of neural networks via their effects upon astrocytes (Grimm et al., 2024). This year compelling data from a range of species (including *Drosophila* and zebrafish, in addition to mammals) have shown that astrocytes are "active players in neuromodulation" A (Eroglu, 2025). Much remains to be discovered about the role of neuromodulators, astrocytes and brain energy regulation. But there is evidence from various species and neural networks that astrocytes influence behaviour (Chen et al., 2025, Guttenplan et al., 2025, Lefton et al., 2025, Ma et al., 2016, Mu et al., 2019). For recent reviews on this exciting topic, see Murphy-Royal et al. (2023) and Eroglu (2025).

The way in which the amplitude of the calcium event in astrocytes encodes information (Mu et al. 2019, Tran et al., 2018) suggests there is greater complexity in the calcium activity than is detected with current methods. The Hirase lab turned their attention to cyclic adenosine monophosphate (cAMP) and developed an optogenetic approach to raise cAMP selectively in astrocytes. Triggering production of cAMP causes arteriolar dilations, independent of intracellular calcium (Vittani et al., 2025). Irrespective of inter lab differences in data, there is evidence that astrocytes facilitate homeostasis depending on metabolic needs. The work of Vittani et al. may resolve the controversy of astrocytic calcium and neurovascular coupling. Further work to see how this mechanism is affected by hypoxia and hypercapnia would be of interest. A summary of vasoactive mediators is given in Table 1.

## Non-invasive methods to measure brain $O_2$

If astrocytes are essential for $O_2$ homeostasis one should consider the methods to measure brain $O_2$ and their relative merits. MRI methods to study brain oxygen include susceptibility-based oximetry (Wehrli, 2024), quantitative susceptibility mapping (QSM) (Kudo et al., 2015), $T_2$-relaxation-under-spin-tagging (TRUST) (Liu et al., 2016) and quantitative blood oxygenation level dependent (BOLD) imaging (He & Yablonskiy, 2007) among others. These methods do not require ionising radiation or contrast agents and have great value in epidemiological studies to assess how cerebral perfusion changes during ageing and how this influences $O_2$ availability to the brain (McFadden et al., 2025). Modelling work has incorporated data from across the spectrum of clinical and preclinical research to describe how the respiratory gases enter and leave the brain

**Table 1. Vasoactive mediators in brain blood flow regulation**

| Mediator | Primary source | Constriction or dilation | Key molecular pathways/receptors |
|---|---|---|---|
| Nitric oxide (NO) | Neurons, astrocytes and endothelial cells | Dilation | cGMP pathway, smooth muscle cells |
| Prostaglandin E2 (PGE2) | Astrocytes (via COX-1) | Dilation | Gs coupled receptors stimulate adenylyl cyclase |
| Epoxyeicosatrienoic acid (EETs) | Endothelial cells | Dilation | Activate BK channels in smooth muscle cells |
| 20-hyroxyeicosatetraenoic acid (20-HETE) | Smooth muscle cells, endothelia, pericytes and astrocytes | Constriction | Inhibition of the BK channels in smooth muscle cells |
| Potassium ($K^+$) | Astrocyte endfeet (efflux via BK channels) | Dilation | Hyperpolarisation of smooth muscle cells |
| ATP | Astrocytes | Constriction | P2X1 receptors expressed by smooth muscle cells |
| ADP | Astrocytes | Dilation | P2Y1 on endothelia which activates eNOS |
| Adenosine | Neurons and astrocytes | Dilation | A2A receptors expressed by smooth muscle cells and endothelia |
| $CO_2$ | All respiring brain cells | Dilation | Sodium–bicarbonate cotransporter 1 expressed by astrocytes |

(Buxton, 2024). Positron emission tomography (PET) has been used to study $O_2$ consumption rate. The tracer fluoromisonidazole (FMISO) is labelled with fluorine-18. FMISO diffuses into cells, and a nitro group is reduced. In well oxygenated tissues the nitro group is oxidised and the molecule diffuses out of the cell. In hypoxic tissues, the nitro group remains reduced and the molecule becomes trapped in the cell emitting positrons (Chakhoyan et al., 2017).

BOLD imaging, pioneered 35 years ago (Ogawa et al., 1990), is able to detect changes in blood oxygenation and cerebral blood volume in T2* weighted gradient echo images. The data are collected rapidly, typically using gradient echo planar imaging (GE-EPI) sequences, favouring temporal resolution over spatial resolution, and activation maps depicting the signal change are presented as coloured overlays on high resolution MRI images of the brain. The signal change detected during periods of heightened neuronal activity arises because the T2* is modified by the proximity of the blood-water protons to oxyhaemoglobin. When the ratio of oxygenated blood in the cerebral vessels increases and conversely the deoxyhaemoglobin decreases, due to an activity-dependent increase in blood flow, the T2* weighted signal is increased. The effect size is small, and researchers could struggle to detect the effect from the noise. To improve the signal to noise ratio, block experiments are performed which comprise repeated episodes of neural activity (a task such as finger tapping or whisker stimulation) and collection of data from multiple blocks. Once these data sets are averaged and combined, the effect can be detected more readily. The enthusiasm for functional MRI using BOLD methods has been huge since this gives an insight into brain activity. But the method detects relative changes in blood $O_2$ and does not give a measure of absolute $O_2$ levels. Near infrared spectroscopy (NIRS) has been used to measure changes in brain $O_2$ of babies (Bale et al., 2020). Using similar technology to a pulse oximeter (application of infrared radiation to the skin surface), the amount of light absorption can be used to derive $O_2$ saturation. The limitation is the depth of light penetration so NIRS cannot be used to sample deep brain regions. Application of NIRS in the adult brain is limited by the thicker skull and greater size of the brain, but this has not stopped its use since it is cheap and highly portable.

## Invasive methods to measure brain $O_2$

Beyond humans lies a range of invasive imaging methods to study $O_2$ in the mammalian brain. Optical imaging spectroscopy via a cranial window in rodents can be used to detect relative changes in the ratio of oxyhaemoglobin and deoxyhaemoglobin from the surface of the brain (Berwick et al., 2008). This method is cheaper and easier than BOLD and also benefits from the ease of access to the rodent brain to allow additional elements such as electrophysiology or optogenetics. $O_2$ sensors implanted into the brain tissue such as Clark electrodes, voltammetry via a carbon fibre electrode (Hosford et al., 2019) or the OxyLite™ sensor (Hosford et al., 2018) are useful for certain applications. Notably they give an absolute

measure of $O_2$ in mmHg. However, insertion of devices causes physical damage to the region of the brain from which $O_2$ is sampled. The main issue with implanted sensors is that the sampling area is restricted to the placement of the sensor. Imaging methods, which are explored in the next section, give spatial and temporal information which can be used to create a map of $O_2$ availability.

The challenge of measuring $O_2$ in absolute values across multiple brain regions was addressed by the introduction of phosphorescence lifetime imaging microscopy (PLIM) developed by Sergei Vinogradov in the laboratory of David Wilson (Vinogradov & Wilson, 1994). Applied by the labs of Boas, Devor, Charpak and Sakadžić, this method has shifted the dial for the understanding of mammalian brain $O_2$ values. For example, as shown in Fig. 2, Sakadžić et al. mapped the partial pressure of $O_2$ across the surface of the mouse brain, illustrating the $O_2$ values that occur in the tissue nearby to arterioles and venules (Sakadžić et al., 2010). These data acquired under anaesthesia give some insight into the variability of $O_2$ available to micro brain regions.

Research from the Lesage lab using the PLIM method in awake mice shows that the partial pressure of $O_2$ in the parenchyma in young and middle-aged animals is $\sim$35 mmHg, whereas this falls to $\sim$25 mmHg in older animals (Moeini et al., 2019). Further work from this group and others shows that exercise elevates brain $O_2$ (Lu et al., 2020, Zhang et al., 2019). In 2024 a paper from the Nedergaard lab using bioluminescence in awake mice confirmed the extent and temporal pattern of hypo-xia in the cortex (Beinlich et al., 2024). This paper illustrates the temporal and spatial variations of $O_2$ and suggests hypoxia occurs due to changes in capillary flow. An optical pH sensor to was used to assess whether hypercapnia and $CO_2$ elimination are coupled with $O_2$ availability. The authors report changes in pH during deliberate hypercapnia (inhalation of 10% $CO_2$); hypoxic pockets are reported in all animals whether awake, mobile or under anaesthesia (Beinlich et al., 2024). Immuno-histochemical markers of hypoxia (pimonidazole) have been reported in the single APP knock-in model of Alzheimer's disease (Korte et al., 2024). In the Tg-SwDI mouse model of Alzheimer's disease (in which amyloid-B plaques aggregate in the microvessels of the brain), the use of carbonic anhydrase inhibitors at a pre-symptomatic stage is shown to reduce the burden of amyloid (Gutiérrez-Jiménez et al., 2024). This suggests that maintaining brain blood flow in the normal range could prevent aggregation of amyloid and prevent hypo-xia. A recent human paper shows that healthy carriers of the APOE-ε4 gene exhibit microvascular changes the authors believe are signs of impaired $O_2$ extraction. When blood flow was compared between young, pre-symptomatic APOE-ε4 and APOE-ε3 carriers, elevated flow was detected in APOE-ε4 carriers; the authors suggest this is evidence of microvascular dysfunction since there is reduced capillary blood volume and abnormal capillary transit time, corresponding with suboptimal $O_2$ update (Aamand et al., 2024). A longitudinal MRI perfusion study undertaken with people suffering from Alzheimer's disease reported reduction in microvascular perfusion, which the authors suggest may limit local $O_2$ (Madsen et al., 2022). A 2025 publication documents blood vessel regression in the brains of mice, primates and humans using longitudinal *in vivo* imaging. The authors conclude that blood vessel regression causes a reduction in neuronal activity due to metabolic dysfunction (Gao et al., 2025). Taken together, there is significant evidence that an $O_2$ deficit occurs as mammals age, and this impacts the energy available to neuronal networks.

## Does reduced cerebral perfusion cause or correlate with dementia?

It is well known that blood flow to the brain declines as people age (Christie et al., 2022) and only some people develop dementia. Despite the evidence outlined above, it remains unclear if low flow is a cause or symptom of developing dementia. With regard to gas homeostasis certain questions arise:

(1) Does hypoxia contribute to pathology such as plaques and neurofibrillary tangles?
(2) Is the impact of hypoxia more severe in poorly perfused brain regions?

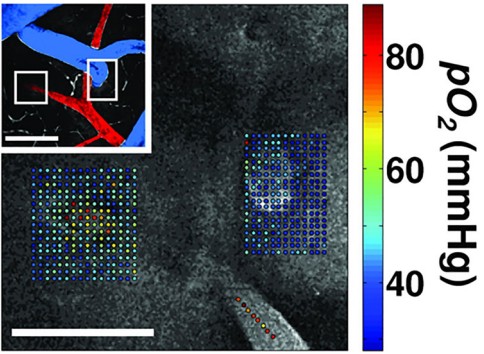

**Figure 2. Detailed maps of oxygen values measured from the rodent brain using phosphorescence lifetime imaging**
Reprinted with permission from Sakadžić et al. (2010): *Nature Methods* 7(9), 755–9. 'Measured pO$_2$ values overlaid with the grey scale phosphorescence intensity image at 40 μm depth. Measurements were performed at locations of descending arteriole (left) and ascending venule (right). Measurement locations were also marked with the white rectangles in the inset (top left), which shows a MIP of 80 μm-thick FITC-labelled microvasculature stack. Artery (red) and vein (blue) are colour-coded for easier identification. Scale bars, 100 μm.'

(3) Does hypoxia or $O_2$ insufficiency contribute to cognitive impairment moment to moment in older people with and without dementia?

Since brain blood flow and the homeostatic regulation of the metabolic gases are in a constant state of flux, one can assume there are moments in health and disease where these mechanisms fail. The evidence from animals shows that regions of the brain become hypoxic even when blood $O_2$ saturation is well within the normal range (Beinlich et al., 2024, Moeini et al., 2019). Exercise is protective against dementia and poor cardiovascular health is a risk factor for dementia (Livingston et al., 2017), while age-related hypertension depresses brain perfusion (Christie et al., 2022). Therefore, it seems the dynamic interaction of the cardiovascular, respiratory and autonomic nervous system, the foundations of gas homeostasis in the brain, should be considered to understand how and when hypoxia becomes problematic.

## Evidence that astrocytes sense changes in $O_2$ tension and cerebral perfusion

A mouse was bred in which a genetically encoded $Ca^{2+}$ sensor, GCaMP3, could be conditionally and selectively expressed within astrocytes (GLAST promotor), and software based on machine learning was used for unbiased selection of active domains (Agarwal et al., 2017). The authors showed that calcium activity occurs in the absence of $IP_3$-dependent release from the endoplasmic reticulum. In fact, they discovered that calcium effluxes from the mitochondria during brief openings of the mitochondrial permeability transition pore (Agarwal et al., 2017). This calcium activity is facilitated by the production of ROS, and sudden increase in ROS production is likened to mitoflashes (Wang et al., 2008). Indeed, the authors found that when ROS production was enhanced by expression of a mutant form of superoxide dismutase 1 ($SOD1^{G93A}$), the microdomain calcium transients were more frequent and had a greater amplitude (Agarwal et al., 2017).

To my knowledge the earliest report that astrocytes detect hypoxia was from the Lauritzen lab (Mathiesen et al., 2012). Lining all cerebral vessels with specialised protrusions called endfeet, astrocytes are well placed to detect changes in $O_2$ availability. Glomus cells are the prototypical $O_2$ sensing cells of the carotid body (see Fig. 3). It has long been known that glomus cells exert a homeostatic role in the systemic circulation, to regulate blood $O_2$ by increasing respiratory drive (Rakoczy & Wyatt, 2018). Research from the Gourine lab *in vitro* experimentally characterised the sensitivity of glomus cells as 37–40 mmHg $O_2$ and the sensitivity of astrocytes as 17 mmHg $O_2$ (Angelova et al., 2015). The threshold of sensitivity for both cell types is close to the boundary between hypoxia and normoxia for the tissue locations in mammals (see Fig. 3). The evidence positions astrocytes as both sensors and the effectors regarding $O_2$ homeostasis in the central nervous system. Mitochondria within astrocytes accumulate nitrite ($NO_2^-$) *in vitro* (Christie et al, 2023) and should this occur *in vivo*, nitrite can be absorbed from the systemic circulation. During hypoxia it is proposed nitrite is reduced to nitric oxide (NO) by the enzyme sulfite oxidase (Christie et al., 2023). Once NO

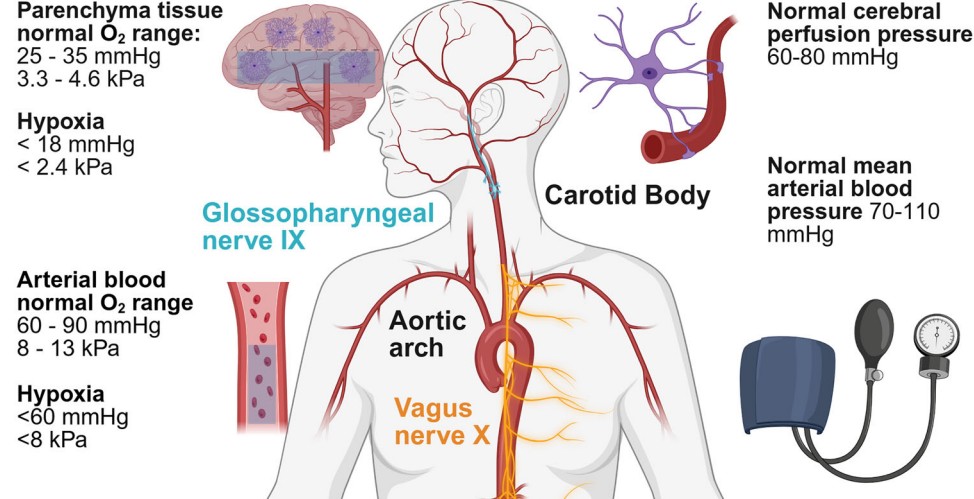

**Figure 3. Summary of the anatomy of peripheral and central mechanisms for the detection of $O_2$ and blood pressure**

The aortic arch is where baroreceptors sense blood pressure, the carotid body is where glomus cells reside which sense $O_2$. The blue regions highlighted in the brain and blood stream indicate hypoxia, which is described numerically on the left of the images. Created in BioRender. Christie, I. (2025) https://BioRender.com/w60e748.

reaches the smooth muscle cells of the cerebral arterioles, the diameter of blood vessels is widened, and a larger blood volume reaches the hypoxic tissue. The mechanism is efficient because $O_2$ is not used to generate NO, as is the case for enzymatic production of NO by endothelia and neurons. When rats were supplemented with exogenous nitrite, the amplitude of hypoxic evoked vasodilations was increased (Christie et al., 2023). Furthermore, there is ample evidence from human studies that increased dietary nitrate can lower blood pressure and boost athletic performance (Jones, 2014).

To understand the role of astrocytes in gas exchange, it is important to consider their role in sensing and controlling cerebral perfusion pressure (CPP), irrespective of classical neurovascular coupling. Figure 3 briefly summarises the anatomy of the peripheral sensors (carotid body, baroreceptors in the aortic arch), cranial nerves and normal and abnormal ranges for $O_2$ and CPP. Evidence from the Filosa lab studying arterioles *in vitro* showed than when the pressure or flow rate was modified artificially, astrocytes responded, which they referred to as flow/pressure-evoked arteriolar constrictions (Kim et al., 2015). A paper from the Gourine lab showed that astrocytes sense CPP and trigger an increase in sympathetic drive when CPP falls (Marina et al., 2020). Blockade of calcium-dependent vesicular signalling mechanisms in astrocytes of the ventrolateral medulla close to the neuronal sympathetic control circuits impaired this response (Marina et al., 2020). When gliotransmission is blocked, the anaesthetised rat is unable to mount an increase in heart rate and raise arterial blood pressure to stabilise the fall in CPP. This work suggests that astrocytes exert a homeostatic influence in the brain with regard to the integration of cardiac output with brain blood flow (Marina et al., 2020).

By sensing and controlling the pressure and rate of blood delivery to the brain, astrocytes influence the amount of $O_2$ available to the brain. So this evidence adds to a theory of astrocyte complexity in which they both sense and regulate $O_2$ and CPP. While this may not be the consensus view, it is possible to interpret these results in the context of a hypothesis termed 'the selfish brain hypothesis of hypertension' (Warnert et al., 2016). The hypothesis states that hypertension occurs because the brain fails to accurately detect (or report) cerebral perfusion within the CNS. Consequently, sympathetic drive is increased to compensate for a falsely perceived hypotension in the brain and the impact on system-wide blood pressure continues to worsen. If astrocytes are the sensors of CPP and brain $O_2$, astrocytes may be a valuable avenue of investigation to understand the fundamental physiology and treatment options for age-related hypertension.

So what happens when the brain actually is hypotensive? People faint and gravity helps to restore brain blood flow since people usually fall to the ground. Fainting is caused by a failure of autoregulation, the mechanisms that ordinarily ensure the brain receives adequate blood supply in the face of sudden changes in blood pressure, posture or other factors (Willie et al., 2014). Autoregulation protects the mammalian brain from hypo- or hyperperfusion, which is reviewed in depth elsewhere (Willie et al., 2014). Lassen's curve describes a broad range of systemic blood pressure (mean ∼60–150 mmHg) which can be tolerated by humans due to autoregulation (Willie et al., 2014). However more recent data suggest this phenomenon is not as robust as once believed, and the dogma of Lassen's curve is being challenged by more recent evidence acquired with more rigorous methods (Brassard et al., 2021). While this review is about $O_2$ homeostasis, it is challenging to consider $O_2$ in isolation, since $CO_2$, pH, respiratory and cardiovascular control mechanisms are interrelated. Hopefully this review highlights the central role of astrocytes in maintaining overall body homeostasis. The brain's respiratory centres, influenced by astrocytic chemosensitivity, stimulate adjustments in breathing rate and depth to return $CO_2$ levels to normal, thereby regulating acid–base balance (Gourine & Dale, 2022). This demonstrates how local astrocytic sensing of metabolic byproducts directly feeds into global physiological regulation, ensuring the brain's stability and survival.

## Impact of Ischaemia and Hypoxia

Astrocytes play a complex and often dual role in ischaemic brain damage. During acute asphyxia, which causes extreme anoxia and hypercapnia, brain injury occurs. This is characterised by energy failure, membrane depolarization and cellular oedema, leading to an increased release and accumulation of excitatory neurotransmitters like glutamate. This causes a massive influx of cellular $Ca^{2+}$ triggering neurotoxic cascades, oxidative stress, inflammation and cell death (Rossi et al., 2007). Astrocyte glycogen stores, while capable of defending against hypoglycaemic damage by providing lactate to neurons, can paradoxically aggravate ischaemic damage in severe conditions due to enhanced lactic acidosis if glycolysis proceeds without $O_2$. Hyperglycaemia during stroke, for instance, correlates with adverse outcomes and exacerbates ischaemic brain damage by increasing lactic acid production (Rossi et al., 2007).

Astrocytic signalling mechanisms also significantly influence neuronal death during ischaemia. Astrocytes can release neurotransmitters such as glutamate, D-serine, ATP and adenosine, all of which have known roles in ischaemic brain damage. For example, ischaemia raises cytoplasmic $Ca^{2+}$ in astrocytes, likely inducing the release of ATP. This ATP can then cause glutamate release via P2X receptors, which is damaging due to

high $Ca^{2+}$ permeability. Conversely, ATP is rapidly converted to adenosine, which is neuroprotective by suppressing excitatory postsynaptic currents. Similarly, while astrocytic glutamate transporters (GLT-1) protect neurons by taking up excess glutamate in early ischaemia, their reversal in prolonged ischaemia can contribute to glutamate release, exacerbating excitotoxicity. Furthermore, brain ischaemia causes astrocytes to swell, leading to the opening of volume-regulated anion channels (VRACs) that efflux glutamate, contributing to excitotoxic damage. The complex role of astrocytic gap junctions is also evident: they can spread either toxic molecules (bystander effect) or protective signals, depending on the context of ischaemia (Rossi et al., 2007).

### Therapeutic potential and emerging avenues

The evolving understanding of astrocytes as central orchestrators of brain gas exchange and $O_2$ homeostasis opens promising avenues for therapeutic interventions in various neurological disorders. Given astrocytes' multifaceted roles, targeting astrocytic functions represents a potent strategy for neuroprotection and disease modification. The direct $O_2$ and perfusion sensing capabilities of astrocytes present unique therapeutic targets. Manipulating astrocytic $Ca^{2+}$ signalling or ATP release could potentially enhance respiratory drive or improve cerebral blood flow in conditions of hypo-perfusion or hypoxia (Marina et al., 2020). For instance, targeting TRPA1 channels in brainstem astrocytes, which contribute to central respiratory $O_2$ sensing, is an emerging area (SheikBahaei, 2020). Furthermore, interventions aimed at restoring proper astrocytic function in the neurovascular unit could ameliorate impaired neurovascular coupling seen in conditions like Alzheimer's disease and stroke, thereby improving $O_2$ and nutrient supply to vulnerable neuronal populations.

The understanding that metabolic alterations influence neurodegenerative disorders, and that hypothalamic astrocytes are pivotal for whole-body energy homeostasis, suggests that targeting astrocytic pathways could lead to new therapeutic strategies for diseases like obesity and associated neurodegeneration.

### Does gas homeostasis impact cognition?

A publication from the Lauritzen lab correlated activity-dependent increases in local $O_2$ consumption with postsynaptic currents (Mathiesen et al., 2011). It is logical to conclude that heightened neuronal activity causes heightened $O_2$ consumption. One can consider a metaphor where the brain is a computer and $O_2$ is the electricity. When a person faints, consciousness rapidly dissipates, as if the plug has been pulled. One can speculate that either there is insufficient $O_2$, or too much $CO_2$ or the sudden shift in pressure–volume compartments of the brain fundamentally disrupts neural computation. It is widely believed that insufficient $O_2$ delivery to the brain occurs during the sudden and temporary loss of consciousness commonly referred to as fainting (Kapoor, 2000). Moving forward from the thought experiment about fainting, I suggest more consideration should be given to $O_2$ insufficiency in age-related cognitive impairment and dementia. Since brain blood flow falls as people age (Christie et al., 2022) and reduced brain perfusion is an early biomarker of dementia onset (Iturria-Medina et al., 2016), it seems plausible that low $O_2$ availability impairs everyday cognition. The regression of cerebral vessels is associated with reduced neuronal activity (Gao et al., 2025). Evidence from animal research suggest that the spatial variation in $O_2$ tension in the brain tissue is spread over 80–10 mmHg depending on proximity to major arteries and activity levels (Sakadžić et al., 2010). Understanding the hypoxic range in mammals is an area of active research (Beinlich et al., 2024). The integrative control of cerebral perfusion is complex, but crucially it is *not static* (Willie et al., 2014). With so many interdependent factors (gases, pH, resistance, pressure, vasoactive signalling molecules) and cell types it is hard to isolate one factor from another. For ageing mammals, reduced perfusion will occur, but the causes could be highly varied and the impact on gas homeostasis may be adaptive.

### Conclusion

Overall, the shift from a neuron-centric view to one that recognizes astrocytes as active, critical players in brain homeostasis provides a vast landscape for developing innovative diagnostic and therapeutic strategies for a wide range of neurological conditions. This review has described how $O_2$ delivery to the brain is intrinsically linked to (1) the lungs, (2) the rate of cerebral perfusion, (3) the metabolic activity of the respiring neurons, (4) the removal of carbon dioxide, and (5) pH regulation. Astrocytes are involved in all aspects of $O_2$ homeostasis and orchestrate the homeostasis of the metabolic environment for neurons. Astrocytes detect changes in $O_2$ (Angelova et al., 2015, Mathiesen et al., 2012) and sense changes in cerebral perfusion pressure (Kim et al., 2015, Marina et al., 2020) and exert bidirectional control over vessel diameter depending on the oxygenation of the tissue (Gordon et al., 2008). When $O_2$ falls below a critical level ($<17$ mmHg) astrocytes release NO, a highly potent vasodilator (Christie et al., 2023). Therefore, in addition to the fundamental role of astrocytes in breathing (Gourine & Dale, 2022), can we conclude that astrocytes are necessary to restore consciousness after fainting? This is a big and whimsical leap. However, if $O_2$ is necessary

for consciousness and astrocytes are the master regulators that govern $O_2$ homeostasis, I invite you to speculate. Neurons are clearly essential for consciousness, but the importance of astrocytes may be more significant than previously realised.

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

## Additional information

### Competing interests

The author declares no competing interests.

### Author contributions

Sole author.

### Funding

This work is funded by a Wellcome Trust Career Development Award.

### Keywords

astrocyte, brain, gas exchange, homeostasis, hypoxia, oxygen transport, perfusion

## Supporting information

Additional supporting information can be found online in the Supporting Information section at the end of the HTML view of the article. Supporting information files available:

**Peer Review History**

