## [Peer Review History · The Journal of Physiology]

Astrocytes: Orchestrators of Brain Gas Exchange and Oxygen Homeostasis

Isabel N Christie

DOI: 10.1113/JP288934

Corresponding author(s): Isabel Christie (i.christie@sheffield.ac.uk)

The following individual(s) involved in review of this submission have agreed to reveal their identity: Barbara Lykke Lind (Referee #1)

Review Timeline:

Submission Date:	24-Mar-2025
Editorial Decision:	07-May-2025
Revision Received:	03-Jul-2025
Accepted:	23-Jul-2025

Senior Editor: Laura Bennet

Reviewing Editor: Nathan Schoppa

Transaction Report:

Dear Dr Christie,

Re: JP-TR-2025-288934 " **Astrocytes: fundamental physiology of oxygen sensing, carbon dioxide elimination, cerebral perfusion and cognition.** " by Isabel N Christie

Thank you for submitting your manuscript to The Journal of Physiology. It has been assessed by a Reviewing Editor and by 2 expert referee(s) and we are pleased to tell you that it is potentially acceptable for publication following satisfactory major revision.

Please address all the points raised and incorporate all requested revisions or explain in your Response to Referees why a change has not been made. We hope you will find the comments helpful and that you will be able to return your revised manuscript within 2 months in order to keep the Special issue on track. If you require longer than this, please contact journal staff: jp@physoc.org. Please note that this letter does not constitute a guarantee for acceptance of your revised manuscript.

ABSTRACT FIGURES: Authors are expected to use The Journal's premium BioRender account to create/redraw their Abstract Figures. Information on how to access this account is here:

<https://physoc.onlinelibrary.wiley.com/journal/14697793/biorender-access>.

REVISION CHECKLIST:

IMPORTANT POINTS TO NOTE WHEN REVISING YOUR MANUSCRIPT:

We look forward to receiving your revised submission.

Yours sincerely,

Laura Bennet
Senior Editor
The Journal of Physiology

REQUIRED ITEMS

- Please include an Abstract Figure file, as well as the Figure Legend text within the main article file. The Abstract Figure is a piece of artwork designed to give readers an immediate understanding of the Review Article and should summarise the main conclusions. If possible, the image should be easily 'readable' from left to right or top to bottom. It should show the physiological relevance of the Review so readers can assess the importance and content of the article. Abstract Figures should not merely recapitulate other figures in the Review. Please try to keep the diagram as simple as possible and without superfluous information that may distract from the main conclusion of the Review. Abstract Figures must be provided by authors no later than the revised manuscript stage and should be uploaded as a separate file during online submission labelled as File Type 'Abstract Figure'. Please ensure that you include the figure legend in the main article file. All Abstract Figures will be sent to a professional illustrator for redrawing and you may be asked to approve the redrawn figure before your paper is accepted.

- Your MS must include a complete "Additional information section" with the following 4 headings and content:

Competing Interests: A statement regarding competing interests. If there are no competing interests, a statement to this effect must be included. All authors should disclose any conflict of interest in accordance with journal policy.

Author contributions: Each author should take responsibility for a particular section of the study and have contributed to writing the paper. Acquisition of funding, administrative support or the collection of data alone does not justify authorship; these contributions to the study should be listed in the Acknowledgements. Additional information such as 'X and Y have contributed equally to this work' may be added as a footnote on the title page.

It must be stated that all authors approved the final version of the manuscript and that all persons designated as authors qualify for authorship, and all those who qualify for authorship are listed.

Funding: Authors must indicate all sources of funding, including grant numbers. If authors have not received funding, this must be stated.

It is the responsibility of authors funded by RCUK to adhere to their policy regarding funding sources and underlying research material. The policy requires funding information to be included within the acknowledgement section of a paper. Guidance on how to acknowledge funding information is provided by the Research Information Network. The policy also requires all research papers, if applicable, to include a statement on how any underlying research materials, such as data, samples or models, can be accessed. However, the policy does not require that the data must be made open. If there are considered to be good or compelling reasons to protect access to the data, for example commercial confidentiality or legitimate sensitivities around data derived from potentially identifiable human participants, these should be included in the statement.

Acknowledgements: Acknowledgements should be the minimum consistent with courtesy. The wording of acknowledgements of scientific assistance or advice must have been seen and approved by the persons concerned. This section should not include details of funding.

- Please upload separate high quality figure files via the submission form.

- Author profile(s) must be uploaded via the submission form. Authors should submit a short biography (no more than 100 words for one author or 150 words in total for two authors) and a portrait photograph of the two leading authors on the paper. These should be uploaded and clearly labelled together in a Word document with the revised version of the manuscript. Any standard image format for the photograph is acceptable, but the resolution should be at least 300 DPI and

preferably more. A group photograph of all authors is also acceptable, providing the biography for the whole group does not exceed 150 words.

- Please include a full title page as part of your main article (Word) file, which should contain the following: title, authors, affiliations, corresponding author name and contact details, keywords, and running title.

- The corresponding author must provide an institutional email address (not a personal address) for their author account. We encourage ALL co-authors to also provide institutional email addresses. If this cannot be provided (as corresponding author), then a stamped letter must be provided from the institution which confirms their role and employment there (please upload this with the revised submission).

EDITOR COMMENTS

We appreciate the manuscript that the author has submitted that reviews the role of astrocytes in maintaining gas homeostasis in the brain. The manuscript has been examined by two expert referees, who felt that the work is comprehensive and explores some novel theses. Thus, it has the potential to be an influential addition to the literature. However, both referees raised a number of significant concerns, all of which will need to be addressed in a revised manuscript that will need to be re-reviewed. Amongst the concerns raised were:

1. As requested by both referees, the review would benefit from further clarification of the main theses and arguments being proposed. The review would also be clearer if it had a more structured organization and potentially a figure that summarizes the author's main argument (as suggested by Referee 1).
2. The review is complex and might benefit from some simplification by removing/reducing some sections. Potential topics to be de-emphasized include some of the historical, foundation content (e.g., Haldane) and the role of hypertension.
3. Both reviewers had specific requests for additions to the review. For example, Referee 2 thought that there should be more critical evaluation of some of methods used to assess oxygenation of the brain.
4. The author should make clearer the distinction between mechanistic points that are clearly supported by the literature and speculative discussion.

REFEREE COMMENTS

Referee #1:

In the review "Astrocytes: fundamental physiology of oxygen sensing, carbon dioxide elimination, cerebral perfusion and cognition", I. Christie has provided an interesting perspective on the astrocyte participation in cerebral blood flow regulation. The review is well written and addresses several highly relevant points in the impact of astrocyte on brain homeostasis. It proposes that the basis of dementia and age-related cognitive decline could be based on a prolonged insufficiency of oxygen delivery to the brain, and it hints that astrocyte functionality could be implicated in this hypoxia. This is an important issue and a complex matter to address over a few pages. The structure of the paper is not all together good, and some work needs to be done in assisting the reader follow the argumentation. The weak storyline may be because the argumentation moves in many directions. I think it is correct to include these several different factors influencing the oxygen level changes, but it is necessary with some clearer explanations for the logic, than what is now provided.

Specific comments:

The title is very general. This loose style does not help the reader in following the logic of the review. The author should consider stating the final message of the paper more boldly and maybe in the title.

The abstract starts with the same sentence as in line 57 of the manuscript. That should be modified. Maybe the last sentence of the abstract should also be reconsidered or incorporated better.

The review contains two figures which are clear in their delivery of this part of the content. But the review would strengthen from an additional simple figure outlining the content of the argumentation. (With arrows meaning regulating, it would look something like: Ast activity → Breathing → oxygen in the brain + Ast activity → Cerebral blood flow → oxygen in the brain). And it could contain more points than what I briefly outline here, of course.

The first section clearly describes how breathing regulates blood gases and how astrocytes regulate breathing. The second section is more complicated to read as it goes through the following 4 topics: 1. How to study astrocytes via intracellular Ca²⁺ levels; 2. What the astrocyte Ca²⁺ levels depend on; 3. That astrocytes contribute to neuronal network activity noradrenaline (NA) dependent; 4 That NA acts as a vasoconstrictor. These diverse topics are presented under the very general title "Capturing data from astrocytes". A more precise title could help prepare the reader for the section.

Topic 2 in this section (from line 90) seems to have some breaches in the flow of argumentation. The author goes from describing how astrocyte Ca²⁺ levels can be modified to addressing the issue of whether astrocyte contributes to functional hyperemia. Firstly, it should be made very clear how this phenomenon differentiates from the regulation of general cerebral blood flow. Secondly, the author wrongly concludes that the papers identifying fast astrocytic Ca²⁺ that occurs on a time scale relevant for functional hyperemia recruitment, were only done with bulk-loaded Ca²⁺ indicators. This is not the case in two of the papers referred to in this discussion (Otsu et al 2015 and Lind et al 2018). The author should maybe also consider to include more recent in vivo investigations showing astrocyte influence on the duration of the functional hyperemia response described in Institoris et al 2022. Thirdly, the bridge from functional hyperemia to the next section is missing. I assume the author argues that when astrocytes are not involved in functional hyperemia, it reflects that they do not respond directly to synaptic activity. Why else would the author then describe how astrocyte Ca²⁺ activity is dependent on noradrenaline and the buildup of ROS species? In short, it would be good if the author could outline the path of argumentation here.

In topics 3 and 4, the author describes how astrocytes NA dependently modify neuronal network activity and then that NA is vasoconstrictive. The review might benefit from also taking in to consideration the recent description of astrocyte dependence on NA shown in Reiman et al 2023. The rationale for including the NA observation is not well explained. However, it poses a very important point for the larger picture of the decline in cerebral NA levels with aging. As it reads now, it seems to be a bit fragmented. The relevance of these findings should be made clearer.

The third section is a very thorough description of how oxygen is measured in the brain and how a slight hypoxia sets in with aging and dementia. This part is good, but maybe a bit long when considering the dimensions of the other text sections. This is only an observation with regard to the balance of the review.

The fourth section is also a bit disorganized and could benefit from a better flow in the organization. I understand the temptation to include hypertension in the story, as it also covers hypotension, but it does not seem to naturally belong in this review. Please explain better why it is included.

Referee #2:

This comprehensive review by Isabel Christie explores the pivotal role of astrocytes in maintaining gas homeostasis in the brain, particularly through sensing and responding to oxygen (O₂) and carbon dioxide (CO₂) levels. The review discusses astrocytes as active regulators of cerebral perfusion and neuronal metabolism, focusing on their roles in functional hyperemia, autoregulation, and the coordination of systemic responses via sympathetic activation. Key findings include evidence that astrocytes sense brain O₂ tension and cerebral perfusion pressure (CPP), using mechanisms such as mitochondrial nitrite reduction to nitric oxide. The manuscript integrates data from animal models and imaging technologies to argue that astrocytic failure in gas sensing contributes to cognitive decline and potentially dementia. Emphasizing astrocytes' dual role as sensors and effectors, the author proposes that astrocytic dysfunction may underlie age-related

hypoperfusion and neurodegenerative changes, offering a compelling argument to reframe our understanding of brain energetics and consciousness.

Major Critique:

1. While rich in content, the review occasionally lacks a clear narrative throughline. A more structured organization (e.g., sectioning into "Astrocyte Physiology," "O₂ Sensing," "Clinical Relevance") could improve readability.
2. Some historical and foundational content (e.g., Haldane, Loeschke) is extensively described, which may detract from space better used to highlight emerging findings or therapeutic implications.
3. The conclusion that astrocytes may be necessary for consciousness is thought-provoking but speculative; stronger framing as a hypothesis or call for further research would be more appropriate.
4. The manuscript mixes high-level mechanistic insights with speculative interpretations without always distinguishing the strength of evidence.
5. While animal studies are emphasized, the relevance to human pathology (e.g., dementia, aging) could be more critically evaluated, especially in light of translational limitations.
6. Pros and Cons of the different methods assessing brain oxygenation/hypoxia can be more elaborated. For example that Clark electrodes and relatives do not only cause physical damage during implantation but also are limited in their spatial readout and thus in the size of the detection area. Similar for PLIM description, here a critical assessment of the pros and cons of the method is lacking (either high spatial or temporal resolution but not both, toxicity of the injected dye).
7. The title promises a discussion of O₂ and CO₂, however most of the manuscript focuses on O₂, while the CO₂ section is very brief. Discussion of CO₂ sensing pathways can be enhanced or the title adjusted accordingly.
8. Regarding the pH measurements in the Beinlich et al paper, the author might have misunderstood that in that particular study pH measurements were solely used to determine whether the oxygen sensor is pH dependent, but has not been used to assess pH dynamics in the cerebral cortex, even though it is a missing piece of puzzle.

Minor Critique:

- Replace: "may be more significant than previously realised" → "may be more significant than previously recognized" (standardize regional spelling or clarify for international audience).
- Consistently format abbreviations on first use (e.g., cerebral perfusion pressure (CPP), oxygen (O₂)).
- "microdomain calcium transients were more frequent/numerous/bigger" → revise for clarity, e.g., "more frequent, larger, and more numerous".
- Review for comma splices and run-on sentences, especially in the more speculative passages.
- Add figure references in-text for Figures 1 and 2, to guide the reader.
- Review for phrases such as "This year", in most cases they refer to 2024 instead of 2025.
- Figure 2: Informative but lacks a caption with physiological interpretation. Please expand the caption to explain what the hypoxia threshold values mean and how they were derived.

END OF COMMENTS

Response to Referee

EDITOR COMMENTS

We appreciate the manuscript that the author has submitted that reviews the role of astrocytes in maintaining gas homeostasis in the brain. The manuscript has been examined by two expert referees, who felt that the work is comprehensive and explores some novel theses. Thus, it has the potential to be an influential addition to the literature. However, both referees raised a number of significant concerns, all of which will need to be addressed in a revised manuscript that will need to be re-reviewed. Amongst the concerns raised were:

1. As requested by both referees, the review would benefit from further clarification of the main theses and arguments being proposed. The review would also be clearer if it had a more structured organization and potentially a figure that summarizes the author's main argument (as suggested by Referee 1).

In response to this comment, I have created a third figure which is actually referred to as Figure 1. This simple flow chart illustrates the main argument and follows the suggestion of Referee 1.

2. The review is complex and might benefit from some simplification by removing/reducing some sections. Potential topics to be de-emphasized include some of the historical, foundation content (e.g., Haldane) and the role of hypertension.

In response to this feedback, I have removed sections about noradrenaline, to keep the review focused on homeostasis of oxygen and astrocytes. I have shortened the historical opening section and shortened the sections on hypertension.

3. Both reviewers had specific requests for additions to the review. For example, Referee 2 thought that there should be more critical evaluation of some of methods used to assess oxygenation of the brain.

I have added more critical evaluation of methods used to assess oxygen in the brain in response the reviewers requests, for example the addition of PET and other methods.

4. The author should make clearer the distinction between mechanistic points that are clearly supported by the literature and speculative discussion.

I have made this distinction clearer by introducing the terms 'thought experiment'. I have made it clear that I invite reader to speculate.

REFEREE COMMENTS

Referee #1:

In the review "Astrocytes: fundamental physiology of oxygen sensing, carbon dioxide elimination, cerebral perfusion and cognition", I. Christie has provided an interesting perspective on the astrocyte participation in cerebral blood flow regulation. The review is well written and addresses several highly relevant points in the impact of astrocyte on brain homeostasis. It proposes that the basis of dementia and age-related cognitive decline could be based on a prolonged insufficiency of oxygen delivery to the brain, and it hints that astrocyte functionality could be implicated in this hypoxia. This is an important issue and a complex matter to address over a few pages. The structure of the paper is not all together good, and some work needs to be done in assisting the reader follow the argumentation. The weak storyline may be because the argumentation moves in many directions. I think it is correct to include these several different factors influencing the oxygen level changes, but it is necessary with some clearer explanations for the logic, than what is now provided.

Specific comments:

The title is very general. This loose style does not help the reader in following the logic of the review. The author should consider stating the final message of the paper more boldly and maybe in the title.

I have changed the title to state the key argument:

Astrocytes: Orchestrators of Brain Gas Exchange and Oxygen Homeostasis

The abstract starts with the same sentence as in line 57 of the manuscript. That should be modified. Maybe the last sentence of the abstract could also be reconsidered or incorporated better.

Thank you for noticing this repetition, I have modified the abstract. I agree the final sentence was poor and this has been re-written.

The review contains two figures which are clear in their delivery of this part of the content. But the review would strengthen from an additional simple figure outlining the content of the argumentation. (With arrows meaning regulating, it would look something like: Ast activity→ Breathing → oxygen in the brain + Ast activity→Cerebral blood flow→oxygen in the brain). And it could contain more points

than what I briefly outline here, of course.

I thank the reviewer for this helpful suggestion. I have created Figure 1 which follows the structure described by referee 1.

The first section clearly describes how breathing regulates blood gases and how astrocytes regulate breathing. The second section is more complicated to read as it goes through the following 4 topics: 1. How to study astrocytes via intracellular Ca²⁺ levels; 2. What the astrocyte Ca²⁺ levels depend on; 3. That astrocytes contribute to neuronal network activity noradrenaline (NA) dependent; 4 That NA acts as a vasoconstrictor. These diverse topics are presented under the very general title "Capturing data from astrocytes". A more precise title could help prepare the reader for the section.

I have modified this section significantly. I have changed the title to: **Highlights of recent astrocyte research**

Topic 2 in this section (from line 90) seems to have some breaches in the flow of argumentation. The author goes from describing how astrocyte Ca²⁺ levels can be modified to addressing the issue of whether astrocyte contributes to functional hyperemia. Firstly, it should be made very clear how this phenomenon differentiates from the regulation of general cerebral blood flow. Secondly, the author wrongly concludes that the papers identifying fast astrocytic Ca²⁺ that occurs on a time scale relevant for functional hyperemia recruitment, were only done with bulk-loaded Ca²⁺ indicators. This is not the case in two of the papers referred to in this discussion (Otsu et al 2015 and Lind et al 2018). The author should maybe also consider to include more recent in vivo investigations showing astrocyte influence on the duration of the functional hyperemia response described in Institoris et al 2022. Thirdly, the bridge from functional hyperemia to the next section is missing. I assume the author argues that when astrocytes are not involved in functional hyperemia, it reflects that they do not respond directly to synaptic activity. Why else would the author then describe how astrocyte Ca²⁺ activity is dependent on noradrenaline and the buildup of ROS species? In short, it would be good if the author could outline the path of argumentation here.

I thank the reviewer for these helpful and constructive comments. I agree there is an oversight regarding the work of Institoris et al and the Gordon lab in general. I have amended this section to include this important contribution and added much more discussion of the work from Grant Gordon including his 2008 Nature paper and his more recent work Tran et al. 2028 Neuron. I have also expanded this section to include recent work from Lind and Volterra 2025, Del Franco et al. 2022. The temporal relationship and the necessity of astrocytic calcium in endfeet and vasodilation is unclear to me at present. I have tried to guide the reader through the

evidence and describe recent evidence in an unbiased and fair way. I have made it clear there is no consensus view at this point in time. I have also created a table of vasoactive substances linked to astrocytes to help make the information as clear as possible for the reader.

In topics 3 and 4, the author describes how astrocytes NA dependently modify neuronal network activity and then that NA is vasoconstrictive. The review might benefit from also taking in to consideration the recent description of astrocyte dependence on NA shown in Reiman et al 2023. The rationale for including the NA observation is not well explained. However, it poses a very important point for the larger picture of the decline in cerebral NA levels with aging. As it reads now, it seems to be a bit fragmented. The relevance of these findings should be made clearer.

I thank the reviewer for this constructive criticism, I agree the rationale for including NA is not well explained and that the section does not read well in its previous form. I have modified this section. I searched for the work of Reiman and was unable to identify the publications suggested by the reviewer. Instead I have cited Carla Eroglu's recent perspective piece in Science (alongside the publication of Chen, Lefton and Guttenplan et al. 2025 all in Science). I have cited Carla's work, because she carefully explains the breakthrough in which is subtle. The triple publication of research papers in Science is a big breakthrough in our understanding of astrocytes. How it relates to oxygen homeostasis is unclear to me at present, but since NA is vasoactive, I believe they could be a meaningful connection.

The third section is a very thorough description of how oxygen is measured in the brain and how a slight hypoxia sets in with aging and dementia. This part is good, but maybe a bit long when considering the dimensions of the other text sections. This is only an observation with regard to the balance of the review.

I thank the reviewer for this constructive comments. I have used more subheadings to guide the reader during this section. I have shortened the description of BOLD since the level of technical detail seemed excessive.

The fourth section is also a bit disorganized and could benefit from a better flow in the organization. I understand the temptation to include hypertension in the story, as it also covers hypotension, but it does not seem to naturally belong in this review. Please explain better why it is included.

I thank the reviewer for this constructive criticism. I have reduced the discussion of hypertension and kept text focused on oxygen.

Referee #2:

This comprehensive review by Isabel Christie explores the pivotal role of astrocytes in maintaining gas homeostasis in the brain, particularly through sensing and responding to oxygen (O₂) and carbon dioxide (CO₂) levels. The review discusses astrocytes as active regulators of cerebral perfusion and neuronal metabolism, focusing on their roles in functional hyperemia, autoregulation, and the coordination of systemic responses via sympathetic activation. Key findings include evidence that astrocytes sense brain O₂ tension and cerebral perfusion pressure (CPP), using mechanisms such as mitochondrial nitrite reduction to nitric oxide. The manuscript integrates data from animal models and imaging technologies to argue that astrocytic failure in gas sensing contributes to cognitive decline and potentially dementia. Emphasizing astrocytes' dual role as sensors and effectors, the author proposes that astrocytic dysfunction may underlie age-related hypoperfusion and neurodegenerative changes, offering a compelling argument to reframe our understanding of brain energetics and consciousness.

Major Critique:

1. While rich in content, the review occasionally lacks a clear narrative throughline. A more structured organization (e.g., sectioning into "Astrocyte Physiology," "O₂ Sensing," "Clinical Relevance") could improve readability.

I thank the reviewer for their time spent critically appraising this article. I have changed the title, used more subheadings and tried to streamline the article to make it more structured.

2. Some historical and foundational content (e.g., Haldane, Loeschke) is extensively described, which may detract from space better used to highlight emerging findings or therapeutic implications.

I thank the reviewer for this feedback and I agree the historical text was excessive. I have reduced the historical foundations, citing other review articles, to preserve word count for more exciting forward looking research.

3. The conclusion that astrocytes may be necessary for consciousness is thought-provoking but speculative; stronger framing as a hypothesis or call for further research would be more appropriate.

I have made it much clearer that this a thought experiment based entirely on speculation. I have explicitly told the reader that I invite them to speculate.

4. The manuscript mixes high-level mechanistic insights with speculative interpretations without always distinguishing the strength of evidence.

I thank the reviewer for pointing this out both to myself and the handling editor. I have tried to make it much more clear in the new draft.

5. While animal studies are emphasized, the relevance to human pathology (e.g., dementia, aging) could be more critically evaluated, especially in light of translational limitations.

My own research spans both animal and human research. In light of this feedback I have tried to incorporate more critical evaluation of human pathology.

6. Pros and Cons of the different methods assessing brain oxygenation/hypoxia can be more elaborated. For example that Clark electrodes and relatives do not only cause physical damage during implantation but also are limited in their spatial readout and thus in the size of the detection area. Similar for PLIM description, here a critical assessment of the pros and cons of the method is lacking (either high spatial or temporal resolution but not both, toxicity of the injected dye).

I agree with the reviewer, I have explained the spatial limitations more clearly in the revised document.

7. The title promises a discussion of O₂ and CO₂, however most of the manuscript focuses on O₂, while the CO₂ section is very brief. Discussion of CO₂ sensing pathways can be enhanced or the title adjusted accordingly.

I wholeheartedly agree with the reviewer and I have modified the title accordingly.

8. Regarding the pH measurements in the Beinlich et al paper, the author might have misunderstood that in that particular study pH measurements were solely used to determine whether the oxygen sensor is pH dependent, but has not been used to assess pH dynamics in the cerebral cortex, even though it is a missing piece of puzzle.

Thank you for making this insight clarification, I had misunderstood the Beinlich et al. paper. I have modified the text accordingly.

Minor Critique:

- Replace: "may be more significant than previously realised" → "may be more significant than previously recognized" (standardize regional spelling or clarify for international audience).

Agreed and amended

- Consistently format abbreviations on first use (e.g., cerebral perfusion pressure (CPP), oxygen (O₂)).

Agreed and amended

- "microdomain calcium transients were more frequent/numerous/bigger" → revise for clarity, e.g., "more frequent, larger, and more numerous".

Agreed and modified

- Review for comma splices and run-on sentences, especially in the more speculative passages.
- Add figure references in-text for Figures 1 and 2, to guide the reader.

Thank you, this is helpful comment and figure references have been added.

- Review for phrases such as "This year", in most cases they refer to 2024 instead of 2025.

Apologies, the review was written in 2024, the text has been amended.

- Figure 2: Informative but lacks a caption with physiological interpretation. Please expand the caption to explain what the hypoxia threshold values mean and how they were derived.

I have given more citations in the text regarding the threshold values.

Dear Dr Christie,

Re: JP-TR-2025-288934R1 "**Astrocytes: Orchestrators of Brain Gas Exchange and Oxygen Homeostasis**" by Isabel N Christie

We are pleased to tell you that your paper has been accepted for publication in The Journal of Physiology.

Authors should note that it is too late at this point to offer corrections prior to proofing. Major corrections at proof stage, such as changes to figures, will be referred to the Editors for approval before they can be incorporated. Only minor changes, such as to style and consistency, should be made at proof stage. Changes that need to be made after proof stage will usually require a formal correction notice.

Yours sincerely,

Laura Bennet
Senior Editor
The Journal of Physiology

P.S. - You can help your research get the attention it deserves! Check out Wiley's free Promotion Guide for best-practice recommendations for promoting your work at www.wileyauthors.com/eoo/guide. You can learn more about Wiley Editing Services which offers professional video, design, and writing services to create shareable video abstracts, infographics, conference posters, lay summaries, and research news stories for your research at www.wileyauthors.com/eoo/promotion.

IMPORTANT NOTICE ABOUT OPEN ACCESS: To assist authors whose funding agencies mandate public access to published research findings sooner than 12 months after publication, The Journal of Physiology allows authors to pay an Open Access (OA) fee to have their papers made freely available immediately on publication.

You can check if your funder or institution has a Wiley Open Access Account here: <https://authorservices.wiley.com/author-resources/Journal-Authors/licensing-and-open-access/open-access/author-compliance-tool.html>.

EDITOR COMMENTS

Reviewing Editor:

Congratulations! Both referees of the original manuscript thought that the author addressed prior concerns quite well and that the work represents an important contribution. The topical review is now acceptable for publication.

REFeree COMMENTS

Referee #1:

I find the manuscript improved by the adjustments and would like to support that it is accepted for publication.

Referee #2:

The author has responded well to the critique. The review is by now a great contribution to the field. I have no additional critique